# Sex-Specific Variations in Glycated Hemoglobin Responses to Structured Exercise in Type 2 Diabetes: Healthcare Implications of Walking and Strength Training on Glycemic Control

**DOI:** 10.3390/healthcare12151528

**Published:** 2024-08-01

**Authors:** Alexis C. King, Cynthia Villalobos, Paul Vosti, Courtney D. Jensen

**Affiliations:** 1Department of Health and Exercise Sciences, University of the Pacific, Stockton, CA 95211, USA; 2St. Joseph’s Medical Center, Stockton, CA 95204, USA

**Keywords:** type 2 diabetes mellitus, HbA1c, sex differences, aerobic exercise, resistance training

## Abstract

Type 2 diabetes mellitus (T2DM) affects one in ten individuals in the United States, with rates expected to rise significantly. This novel study aimed to evaluate the impact of a structured exercise program on glycated hemoglobin (HbA1c) levels among males and females with T2DM, and to compare the effects of different volumes of combined aerobic and resistance exercise. A total of 67 adult participants with T2DM were randomly assigned to two groups: Group 1 (exercise classes and walking sessions) and Group 2 (exercise classes only). After 10 weeks, 39 participants completed the intervention and 34 had complete HbA1c records. Results indicated a significant improvement in HbA1c levels overall, with males exhibiting a greater decrease compared to females. Minimal baseline differences were observed between the walking and non-walking groups and improvements in HbA1c were noted in both groups with no significant differences. These findings suggested potential sex-specific differences in response to structured exercise programs. The study highlighted the importance of tailored exercise interventions in healthcare while managing T2DM. Further research is necessary to optimize exercise prescriptions and evaluate long-term benefits, but the current evidence supports structured exercise as a valuable component of comprehensive diabetes care. This research underscores the need for personalized approaches in exercise regimens, contributing to the growing body of knowledge on sex-specific responses to T2DM interventions.

## 1. Introduction

In the United States, one in every ten individuals has diabetes, with 95% of cases being type 2 diabetes mellitus (T2DM) [1]. By 2030, it is estimated that the number of diagnoses could reach 54.9 million Americans [2]. T2DM occurs when pancreatic β-cells fail to produce insulin due primarily to environmental factors. Engaging in regular physical activity can serve as an effective method to prevent, delay, or manage T2DM. The rising rates of obesity and sedentary lifestyles are significant contributors to this epidemic.

The diagnosis of T2DM is attributed to a combination of lifestyle and several biological and environmental factors [3]. Individuals with T2DM often present with psychological difficulties, cardiovascular complications, and low physical activity levels [3,4]. The frequency and severity of T2DM symptoms differ between sexes, with women experiencing a higher risk of depression, elevated cardiovascular morbidity and mortality rates, and poorer exercise capacities [4,5,6]. Globally, there are 12 million more diagnosed cases of T2DM in men than in women [2]. Additionally, there are differences in the demographic and anthropometric profiles of T2DM patients between sexes, with men typically being diagnosed at a younger age and having a lower body mass index (BMI) compared to females [4,6]. Obesity is the primary risk factor for T2DM in both sexes and is often the target of non-pharmacological-based interventions for T2DM [4].

An important measure for managing and diagnosing T2DM is glycated hemoglobin (HbA1c). HbA1c reflects the average blood glucose levels over the past two to three months and is a critical indicator of long-term glycemic control. Lowering HbA1c levels can reduce the risk of diabetes-related complications, making it a key target in T2DM management.

Exercise is commonly prescribed to individuals with T2DM to reduce obesity, increase insulin sensitivity, improve quality of life (QoL), and enhance cardiovascular health [7]. Both the American College of Sports Medicine (ACSM) and American Diabetes Association (ADA) recommend at least 150 min per week of moderate to vigorous aerobic exercise, with a combination of two to three days of resistance training [8]. However, fewer than 40% of individuals with T2DM report engaging in physical activity [9], while others overestimate their level of activity [8]. Research suggests that combining aerobic and resistance training may be more effective than either type alone [10], although there are limitations to prescribing higher volumes of exercise. Additionally, exercise has been shown to effectively prevent and manage T2DM symptoms independent of its impact on BMI [10]. However, the efficacy of exercise interventions in this population varies due to factors such as age, baseline fitness levels, and sex [10,11]. Despite numerous trials examining the effects of exercise on individuals with T2DM, few studies have analyzed the benefits of these interventions exclusive to the participants’ sex [12].

The aim of this study was two-fold: (1) evaluate glycated hemoglobin (HbA1c) changes to structured exercise among males and females with T2DM; (2) evaluate different volumes of combined aerobic and resistance exercise on HbA1c levels. We hypothesized males with T2DM would exhibit improved changes in glycemic control, cardiovascular health parameters, and quality of life outcomes compared to females following participation in structured exercise programs. Additionally, we anticipated that factors such as age, baseline physical fitness, and BMI would moderate the effects of exercise interventions on T2DM outcomes differently between males and females.

## 2. Materials and Methods

We assessed the impact of a 10-week comprehensive exercise regimen, incorporating cardiovascular training, strength training, and flexibility exercises, on adult patients (>18 years) diagnosed with non-insulin-dependent T2DM. The intervention, approved by the institutional review boards of a hospital and collaborating university, aimed to assess changes in HbA1c levels. Participants were randomly assigned to one of two groups: Group 1 attended exercise classes twice a week and engaged in a one-hour walk three times a week, while Group 2 attended exercise classes twice a week without prescribed walking sessions.

### 2.1. Randomization and Outcome Measures

Participants were randomly assigned to groups by drawing tokens from a bag, with 50% of the tokens being blue (Group 1—walking intervention) and the remaining 50% black (Group 2—non-walking control). At baseline and follow-up, all participants underwent comprehensive testing involving demographic and health history, a subjective quality of life assessment, anthropometric evaluation, cardiometabolic risk factor screening, and various physical function tests incorporating both strength and flexibility components. To avoid interrater error, a single clinician conducted all testing at baseline and follow-up (Figure 1).

The anthropometric measurements were BMI and body fat percentage, calculated using the bioelectrical impedance analysis (BIA). The cardiovascular measurements were resting heart rate and blood pressure. Blood pressure was measured using a stethoscope and blood pressure cuff. Hypertension was assessed with a blood pressure reading over 120/80 mmHg or the use of anti-hypertensive medication. HbA1c levels were determined through blood tests conducted at the hospital laboratory; it was analyzed consistently by the same personnel during baseline and follow-up. The functional tests were VO_2_ max, a six-minute walk, timed up-and-go, chair stand, sit-to-stand, arm curl, grip strength, universal machine (UM) push and pull, epic lift, sit-and-reach, functional reach, and back scratch. Quality of life was evaluated using a QoL inventory covering multiple domains of well-being (SF-36).

### 2.2. Exercise Intervention

Exercise classes were conducted biweekly by the clinician responsible for the baseline testing, supported by interns who specialized in group exercises. Each session comprised three segments: the check-in, exercise, and cool-down. The exercise regimen included aerobic, flexibility, and resistance training components, tailored to individual participants by the lead clinician. Aerobic exercises, including treadmill, Nu-step, upper body ergometer (UBE), and recumbent bike, were performed in six-minute intervals, totaling 24 min. Participants were instructed to exert themselves at 60–75% of their maximum heart rate, with those on heart medication targeting a Borg rating of perceived exertion score between 12 and 14. Resistance exercises included chest press, latissimus dorsi pulldowns, triceps extensions, standing rows, bicep curls, wall squats, and step-ups. During the first exercise session, participants performed a single set of ten repetitions on each exercise. During the second exercise session, participants performed two sets of ten repetitions. During each subsequent session, participants performed three sets of ten repetitions.

At the start of each session, participants checked in and retrieved their exercise cards detailing the program and their target heart rate. Blood glucose levels were recorded from their personal devices and blood pressure was measured by the interns overseeing their session. Following aerobic exercise, the heart rate was monitored using a pulse oximeter, with guidance provided to achieve the target exertion level. The cool-down phase included stretching major muscle groups for approximately 45 s each, incorporating yoga poses, and concluding with deep breathing exercises.

Upon the completion of the 10-week program, all participants underwent a re-evaluation, with walking participants submitting written walking logs to document adherence to the prescribed walking regimen. The same clinician who conducted the baseline assessments repeated all tests during the post-evaluation phase.

### 2.3. Statistical Analysis

Descriptive statistics were calculated for the entire sample to characterize participant demographics. Mean differences between the walking and non-walking groups at baseline, as well as between males and females, were examined using independent samples *t*-tests. Baseline differences in categorical data were evaluated using chi-squared tests. A repeated measures ANOVA with the Greenhouse–Geisser correction compared HbA1c levels at baseline and follow-up between males and females. Bonferroni post hoc analyses were conducted to assess specific group differences. All statistical analyses were performed using SPSS version 24 (IBM SPSS Statistics, IBM Corporation, Chicago, IL, USA). Significance was set at two-sided *p* < 0.05.

## 3. Results

A total of 67 participants diagnosed with T2DM were enrolled in the study; of these, 39 successfully completed the intervention and follow-up testing, with 34 having complete HbA1c records (Table 1). The participants’ ages ranged from 39 to 87 years (mean: 68.3 ± 10.7). Most participants were female (61.2%; N = 41). The BMI ranged from 19.2 to 54.5 kg/m^2^ (mean: 32.3 ± 6.7) and body fat percentage ranged from 19.3% to 54.4% (mean: 39.3 ± 6.9). Most of the participants were obese (59.7%) and hypertensive (66.7%) as defined by SBP ≥ 140 and/or DPB ≥ 90; medication use was not considered (Table 1). The non-walking group had 17 males (39.5%) and 26 females (60.5%). The walking group had 9 males (39.1%) and 14 females (60.9%).

Participants were evaluated on three cardiometabolic risk factors: blood pressure (SBP: 128.3 ± 11.9 mmHg, DBP: 75.0 ± 8.4 mmHg), heart rate (78.1 ± 14.3 bpm), and HbA1c (7.0 ± 1.1). Their QoL, assessed through the SF-36 questionnaire, yielded a mean score of 59.8 ± 17.6. The baseline functional assessments revealed a mean timed up-and-go (TUG) test time of 7.3 ± 2.7 s, a mean back scratch distance of −11.9 ± 6.3 inches, and an average grip strength of 59.5 ± 21.3 kg.

### 3.1. Male vs. Female Comparison

Among the 39 participants who completed the exercise program, 34 (13 males and 21 females) had a complete HbA1c records. Sex was not related to trial completion (*p* = 0.660) or baseline HbA1c levels (*p* = 0.117). No component of baseline characteristics differed between completers and non-completers (*p* > 0.05), and there was no significant difference in HbA1c levels with exercise between the two groups (*p* = 0.234). The repeated measures ANOVA indicated an improvement in HbA1c levels between the two groups (F = 7.878; *p* = 0.008) and an interaction effect with sex (F = 6.734; *p* = 0.014) (Table 2). Male participants showed a greater decrease in HbA1c values compared to females (0.61 vs. to 0.02, respectively) (Table 3). The Bonferroni correction revealed a mean difference of 0.316 between pre- and post-HbA1c values among completers (*p* = 0.008).

### 3.2. Walking vs. Non-Walking Intervention

Minimal group differences were observed at baseline. Participants in the non-walking group were slightly older by 4.7 years (*p* = 0.092) and had a lower BMI (3.3 points; *p* = 0.058). There were no significant differences in body fat percent (*p* = 0.507), HbA1c (*p* = 0.512), other cardiometabolic parameters, or other physical functioning assessments. The linear regression analysis found that an elevated baseline body fat percent (*p* = 0.001) and improvements in strength, assessed through arm curls (*p* = 0.009) and grip strength (*p* = 0.042), were associated with poorer outcomes in HbA1c (R^2^ = 0.733; *p* < 0.001). Participants who completed the exercise intervention showed improvements in 13 of 16 assessments (*p* < 0.05), including HbA1c (*p* = 0.045), with no differences observed between exercise groups.

## 4. Discussion

This study had two primary objectives: first, to assess changes in HbA1c levels resulting from structured exercise programs in both males and females with T2DM; secondly, to evaluate the impact of varying volumes of combined aerobic and resistance exercises on HbA1c levels. Our hypothesis regarding sex-specific variations in HbA1c responses to structured exercise interventions among patients with T2DM was partially supported. In the comparison between males and females, male participants exhibited a greater decrease in HbA1c values compared to females, indicating potential sex-specific differences in the response to structured exercise in terms of glycemic control. Regarding the comparison between walking and non-walking interventions, minimal group differences were observed at baseline. While improvements in HbA1c were observed in participants who completed the exercise intervention, regardless of the walking or non-walking program, no significant differences were noted between the two intervention groups.

Given the importance of maintaining healthy HbA1c ranges in managing diabetes and its associated complications within this group, exercise interventions similar to those implemented in our study have shown to enhance various physiological and psychological parameters, thus contributing to an improved QoL [13,14]. These findings underscored the clinical significance of tailoring exercise programs to cater to the specific needs of both male and female individuals at risk for or diagnosed with T2DM. Interestingly, increasing the volume of aerobic exercise did not yield superior outcomes in physical functioning or cardiometabolic factors compared to a non-walking control group. Additionally, we observed a paradoxical association between improvements in strength and poorer HbA1c outcomes; specifically, higher levels of arm curls and grip strength were correlated with elevated HbA1c levels during the post-evaluation assessments. These findings suggested that while combining aerobic and resistance training is effective in enhancing physical functioning and cardiometabolic factors, further research is warranted to understand a precise exercise prescription that optimally assists T2DM participants in controlling HbA1c levels for both sexes.

### 4.1. Sex-Based Differences

The comparison between males and females revealed interesting sex-based differences in response to the exercise intervention. Although there was no significant difference in trial completion rates or baseline HbA1c levels between the sexes, males exhibited a greater reduction in HbA1c levels compared to females. This differential response could be attributed to several factors, including physiological differences in muscle mass and metabolism, as well as potential differences in adherence to the exercise regimen or intensity of physical activity [15,16]. The significant interaction effect of sex on HbA1c improvement, as indicated by the repeated measures ANOVA, suggested that males may derive greater glycemic benefits from exercise. However, it is crucial to consider that the overall mean difference in HbA1c, though statistically significant, may not be clinically significant for all individuals.

### 4.2. Walking vs. Non-Walking Intervention

The comparison between the walking and non-walking interventions did not reveal significant baseline differences in most parameters, except for age and BMI. Despite the minimal differences at baseline, the linear regression analysis identified an elevated baseline body fat percentage and improvements in strength as predictors of poorer HbA1c outcomes. This finding suggested that participants with a higher baseline body fat may face greater challenges in achieving glycemic control, possibly due to insulin resistance associated with higher adiposity [17,18].

### 4.3. Improvements in Physical Function and QoL

The exercise intervention led to significant improvements in 13 of 16 assessments, including HbA1c, physical function, and QoL. The notable improvements in the timed up-and-go (TUG) test, back scratch distance, and grip strength indicated enhanced physical mobility and flexibility, which are crucial for the overall well-being of individuals [19,20]. The improvement in the SF-36 QoL score reflected the positive impact of exercise on participants’ perceived physical and mental health [21].

### 4.4. Strengths, Limitations, and Future Research

The study had several limitations that warrant consideration. The relatively small sample size of 67 participants with T2DM may have limited the generalizability of the findings to the broader population. We must recognize that the 58% dropout rate raised concerns regarding the feasibility of the interventions, although it is notable that 39 participants successfully completed the 10-week program. Furthermore, while the study allocated participants into two groups with different exercise regimens, participants in the exercise and walking group were required to submit written walking logs to track their adherence, thus, introducing the potential for inaccuracies or biases in self-reported data. Lastly, the evaluation of the exercise programs’ effects was limited to a relatively short duration of 10 weeks. This timeframe may not be adequate to identify long-term outcomes or the sustainability of the effects observed. Addressing these limitations in future research endeavors would contribute to a more robust understanding of the implications of exercise interventions in patients with T2DM.

Moving forward, future research should consider the benefits of larger interventions involving multi-center trials with diverse patient populations to enhance the generalizability of results. Strategies to improve intervention adherence and minimize dropout rates also warrant exploration. The idea behind a mixed-methods approach to interview TD2M on the barriers and motivators to exercise could be useful in better implementing exercise-related interventions. Furthermore, employing more objective measures, such as wearable activity trackers, to assess physical activity levels could mitigate potential biases associated with self-reporting. This has been conducted in several studies within T2DM patients [22,23,24]. Longitudinal studies which have extended follow-up periods beyond the 10 weeks evaluated in this study have comprehensively assessed the sustained impact of exercise interventions on glycemic control and other health and QoL outcomes in patients with T2DM [24,25,26]. Finally, investigating the optimal frequency, intensity, duration, and modality of exercise interventions tailored to individual patient characteristics would provide valuable insights into personalized exercise prescriptions for optimizing glycemic control and overall health in this population. Addressing these research directions would advance our understanding of the role of exercise in managing T2DM and inform evidence-based interventions to improve patient outcomes in clinical practice.

## 5. Conclusions

This study demonstrated that structured exercise interventions can significantly improve glycemic control, physical function, and quality of life in individuals with T2DM. The findings highlighted the importance of personalized exercise programs that consider sex-based differences and baseline characteristics, such as body fat percentage, to optimize health outcomes.

Males appeared to benefit more from the exercise intervention in terms of HbA1c reduction compared to females. Additionally, the complexity of the relationship between different types of exercise and glycemic outcomes underscored the need for further research to elucidate the mechanisms behind these effects. The association of a higher baseline body fat percentage with poorer HbA1c outcomes emphasized the need for comprehensive strategies that include both aerobic and resistance training, along with dietary interventions, to manage T2DM effectively.

Overall, this study reinforced the role of regular physical activity as a cornerstone of T2DM management and provided valuable insights for tailoring exercise interventions in healthcare to maximize benefits for diverse patient populations. Future studies should explore the long-term effects of different exercise modalities and intensities, as well as their interactions with other lifestyle factors, to develop more effective and individualized treatment plans for T2DM patients.

## Figures and Tables

**Figure 1 healthcare-12-01528-f001:**
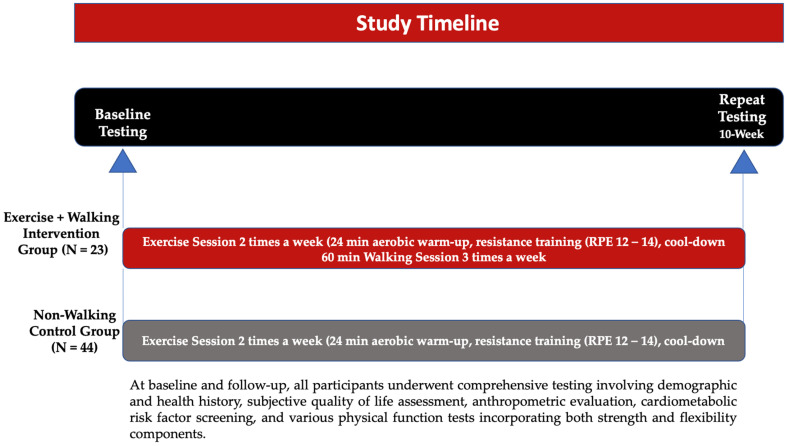
Study timeline.

**Table 1 healthcare-12-01528-t001:** Demographic and anthropometric data, cardiometabolic parameters, and medical history at intake (two-sided significance reported).

Variables	Total Sample (N = 67)	Walking (N = 23)	Non-Walking (N = 44)	*p*	Males (N = 26)	Females (N = 41)	*p*
HbA1c (%)	7.0 ± 1.1	6.9 ± 1.1	7.1 ± 1.2	0.512	7.3 ± 1.4	6.8 ± 1.0	0.117
Age (years)	68.3 ± 10.7	71.4 ± 8.2	66.7 ± 11.6	0.092	69.9 ± 9.8	67.2 ± 11.1	**0.031**
Race (% white)	67.1%	65.2%	68.1%	1.000	78.9%	60.0%	0.145
Hypertension (% yes)	66.7%	77.4%	59.0%	0.104	69.2%	65.0%	0.727
Hyperlipidemia (% yes)	53.8%	56.5%	52.2%	0.790	57.7%	51.2%	0.618
Heart Attack (% yes)	23.8%	17.4%	27.3%	0.562	30.8%	19.5%	0.053
Body Fat (%)	39.3 ± 6.9	38.4 ± 6.6	39.7 ± 7.1	0.507	33.5 ± 6.0	43.0 ± 4.5	**<0.001**
Obesity (% yes)	59.7%	60.9%	59.0%	0.888	65.4%	56.1%	0.458
BMI (kg/m^2)^	32.3 ± 6.7	30.1 ± 5.2	33.3 ± 7.2	0.058	32.5 ± 5.7	32.2 ± 7.4	0.859
SBP (mmHg)	128.3 ± 11.9	126.9 ± 11.0	129.2 ± 12.4	0.469	127.9 ± 14.1	128.5 ± 10.4	0.857
DBP (mmHg)	75.0 ± 8.4	74.9 ± 8.6	75.2 ± 8.4	0.908	74.6 ± 8.8	75.2 ± 8.2	0.777
HR (bpm)	78.0 ± 14.3	80.0 ± 14.0	77.2 ± 14.7	0.465	76.6 ±17.4	79.1 ± 12.1	0.517
Timed up-and-go (s)	8.0 ± 2.7	7.6 ± 1.9	8.2 ± 3.0	0.335	8.9 ± 3.8	7.5 ± 1.5	0.079
Back Scratch (cm)	−11.9 ± 6.3	−11.3 ± 7.2	−12.3 ± 5.8	0.547	−15.4 ± 6.2	−9.7 ± 5.2	**<0.001**
Grip Strength (kg)	59.5 ± 21.3	60.0 ± 21.3	59.6 ± 21.6	0.940	75.8 ± 19.6	49.2 ± 15.0	**<0.001**
QoL Score	59.8 ± 17.6	59.7 ± 17.3	59.8 ± 18.0	0.972	59.9 ± 16.0	59.7 ± 18.6	0.965

Values in bold are representative of a significance value *p* ≤ 0.05.

**Table 2 healthcare-12-01528-t002:** Repeated measures ANOVA with Greenhouse–Geisser correction.

	Type III Sum of Squares	DF	Mean Square	F	Sig
HbA1c Value	1.601	1.000	1.601	7.878	**0.008**
HbA1c Value by Sex	1.369	1.000	1.369	6.734	**0.014**

Values in bold are representative of a significance value *p* ≤ 0.05.

**Table 3 healthcare-12-01528-t003:** Paired samples *t*-tests comparing the pre and post exercise program values of individuals who completed the program.

Variable	Walking			Non-Walking		
	Pre	Post	Two-Sided *p*	Pre	Post	Two-Sided *p*
HbA1c (%)	6.9 ± 1.1	6.4 ± 0.55	**0.23**	7.1 ± 1.2	6.8 ± 0.9	0.313
Body Fat (%)	38.4 ± 6.6	36.6 ± 7.3	0.229	39.6 ± 7.1	38.4 ± 6.5	0.575
BMI (kg/m^2^)	30.1 ± 5.2	29.0 ± 5.3	**0.008**	33.3 ± 7.1	32.5 ± 5.4	0.150
SBP (mmHg)	126.9 ± 11.0	121.9 ± 13.5	0.134	129.1 ± 12.4	127.5 ± 9.0	0.095
DBP (mmHg)	75.0 ± 8.6	70.1 ± 5.9	0.089	75.2 ± 8.4	71.8 ± 7.8	**0.009**
Resting HR (bpm)	80.0 ± 14.0	72.4 ± 10.3	0.090	77.2 ± 14.7	75.4 ± 15.9	0.134
Functional Reach (cm)	10.9 ± 2.6	13.4 ± 3.3	**0.004**	10.5 ± 3.1	11.8 ± 2.6	**0.003**
Arm Curl Amount (kg)	15.3 ± 4.5	20.4 ± 4.9	**<0.001**	13.7 ± 4.5	18.1 ± 3.4	**<0.001**
6 min walk (m)	445.71 ± 79.08	496.4 ± 93.0	**<0.001**	373.1 ± 115.8	411.6 ± 107.2	**<0.001**
Chair Stand	10.7 ± 4.1	15.0 ± 4.2	**<0.001**	10.2 ± 2.7	13.0 ± 2.4	**<0.001**
Sit and Reach (cm)	−4.9 ± 5.7	−1.8 ± 5.4	**0.004**	−5.1 ± 4.2	2.7 ± 4.2	**<0.001**
Timed up-and-go (s)	7.5 ± 1.9	6.5 ± 2.1	**<0.001**	8.2 ± 3.0	7.2 ± 2.3	**<0.001**
Back Scratch (cm)	−11.3 ± 7.2	−8.0 ± 6.7	**0.022**	−12.3 ± 5.8	−9.7 ± 4.3	**<0.001**
Grip Strength (kg)	60.1 ± 21.3	65.2 ± 22.4	0.236	59.6 ± 21.6	61.1 ± 19.6	0.514

Values in bold are representative of a significance value *p* ≤ 0.05.

## Data Availability

The data presented in this study are available on request from the corresponding author due to (specify the reason for the restriction).

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
