# Peer review of "Sex-Specific Variations in Glycated Hemoglobin Responses to Structured Exercise in Type 2 Diabetes: Healthcare Implications of Walking and Strength Training on Glycemic Control"

_healthcare, 2024, doi:10.3390/healthcare12151528_

Round 1
Reviewer 1 Report
Comments and Suggestions for Authors
1. Explain the importance of HbA1C in the introduction section.
2. Present the grouping and characteristics of each group in a table.
3. Section 2.2, line 95, give the details of exercises, times, etc. in a table.
4. For the convenience of the readers, please draw a schematic figure of how to do the design and get the results.
5. Make a list of all existing abbreviations and put them at the beginning of the article.
6. Please put a section titled limitations and future perspective before the conclusion section and improve the discussion more deeply with up-to-date scientific sources.
Author Response
Dear Reviewer 1,
Comments 1: Explain the importance of HbA1C in the introduction section.
Response 1: Thank you for your feedback. Information about HbA1c has been added to the introduction section; lines 45 - 49.
Comments 2: Present the grouping and characteristics of each group in a table.
Response 2: I believe this is addressed in Table 1; Lines 145 - 147.
Comment 3: Section 2.2, line 95, give the details of exercises, times, etc. in a table.
Response 3: Thank you for your suggestion. Unfortunately, due to space constraints, we are unable to provide a detailed table, but we have provided explanation of the protocol within our methodology in lines 102 - 126.
Comments 4: For the convenience of the readers, please draw a schematic figure of how to do the design and get the results.
Response 4: Thank you for your suggestion. Unfortunately, due to space constraints, we will not be able to include a schematic figure in this article. We believe that the detailed explanation provided in the methods section offers a clear understanding of the study design and the process for obtaining results. We appreciate your understanding.
Comment 5: Make a list of all existing abbreviations and put them at the beginning of the article.
Response 5: Thank you for your suggestion. Unfortunately, due to space constraints, we are unable to make an abbreviation list. We do introduce each abbreviation early on in the introduction and methodology for the reader.
Comment 6: Please put a section titled limitations and future perspective before the conclusion section and improve the discussion more deeply with up-to-date scientific sources.
Response 6: Thank you for the recommendation. A section has been added to reflect this request and can be found on lines 234 - 264.
Reviewer 2 Report
Comments and Suggestions for Authors
The manuscript “Sex-Specific Variations in HbA1c Responses to Structured Exercise in Type 2 Diabetes: Healthcare Implications of Walking and Strength Training on Glycemic Control” was submitted by authors. I appreciated the authors to carry out such as nice work. The following suggestions may be consider for improving the manuscript. There few important corrections to be rectified.
Highlight the novelty of study in abstract of part.
“resistance training” is needed as keyword?
Could you include the age details of patients? Are they below 18?
Did you receive the any animal ethical approval for this study?
Author Response
Dear Reviewer 2,
Comment 1: Highlight the novelty of study in abstract of part.
Response 1: thank you for your suggestion. Abstract has been rewritten to include emphases on novelty; Lines 10-12 and 23 - 24.
Comment 2: “resistance training” is needed as keyword?
Response 2: "Resistance training" is a critical keyword for this study as it highlights the specific type of exercise intervention examined.
Comment 3: Could you include the age details of patients? Are they below 18?
Response 3: All participants were over the age of 18 as indicated on line 75. In addition, The participants ages ranged from 39 to 87 years (mean: 68.3 ± 10.7) which can be found on Line 140 – 141.
Comment 4: Did you receive the any animal ethical approval for this study?
Response 4: We did not require animal ethical approval for this study as our study did not involve the participation or use of animals in our research.
Reviewer 3 Report
Comments and Suggestions for Authors
Dear authors,it was a pleasure to read the study entitled: "Sex-Specific Variations in HbA1c Responses to Structured Exercise in Type 2 Diabetes: Healthcare Implications of Walking and Strength Training on Glycemic Control" by Alexis C. King and colleagues with interest. This topic is of some interest and the manuscript is easy to follow.
However, I have the following comments:
1.The manuscript itself was generally well-written, but could also benefit from further editing to remove some expressions that are not used in professional science communication. In general, the article could use some editing by someone who is experienced in scientific communication, particularly in English. Some examples of imprecise language are as follows:
Introduction, lines 47-49. "However, fewer than 40% of individuals with T2DM report engaging in physical activity and more than 20% overestimate their level of activity." Imprecise and clunky writing here.
Results, lines 129-130. "A total of 67 participants diagnosed with T2DM were enrolled in the study, of which 39 successfully completed the intervention and were retained through follow-up testing." Clunky writing.
Results, lines 131-132. "The participants ages ranged from 39 to 87 years." It seems not precise here.
2.Lines 83-84. In this section, anthropometric measurements included BMI and body weight at the same time. Since BMI was calculated from both weight (kg) and height (cm), I strongly advise you to retain BMI and remove the body weight in order to reduce redundancy. In addition, you can also choose to add the information of height (cm).
3.Lines 126-127. Maybe this section should be revised: Statistically significance was set at two-sided P<0.05. In addition, I advise you to retain only two-sided test P values in Table 2.
4.One general comment on results: text often duplicates tables’ content, and the order of your results is a little confusing. I strongly advise you to simplify and adjust this section, as it complicates reading.
5.In Table 1. Hypertension was considered as a major determinant, SBP and DBP were also used to determine whether participants have hypertension. But there are other criteria for hypertension, such as take antihypertensive medications. So I suggest you to explain your diagnostic criteria for hypertension clearly in the Materials and Methods section.
6.Table 2 was not standardized, it is recommended to remove the top and bottom border lines of each row. In addition, I advise you to delete the columns of One-sided P values in Table 2. You can refer to the format of Table 1.
7.I agree with the content of this piece of discussion in general, especially with what is reported and particularly if it was in the introduction. However, there are some problems with it:
Lines 172-175. Such a section is really too repeated with the Introduction, if I can advise, I would delete it. It is also possible to refine and summarize the content of your results.
Furthermore, I suggest supplementing some content of possible mechanisms that could support your results in this study to the second paragraph of the Discussion section.
8.You did no add an appropriate strengths section, I suggest adding it in the Discussion section, which should include at least: a) study the information of male and female respondents separately; b) the participants were monitored comprehensively for 10 weeks, and so on.
Comments on the Quality of English LanguageAuthor Response
Dear Reviewer 3,
Comment 1: Introduction, lines 47-49. "However, fewer than 40% of individuals with T2DM report engaging in physical activity and more than 20% overestimate their level of activity." Imprecise and clunky writing here. Results, lines 129-130. "A total of 67 participants diagnosed with T2DM were enrolled in the study, of which 39 successfully completed the intervention and were retained through follow-up testing." Clunky writing. Results, lines 131-132. "The participants ages ranged from 39 to 87 years." It seems not precise here.
Response 1: Thank you for your edits. We have provided a rewrite of this sentence in lines 54 - 56; 138 - 140;
Comment 2: Lines 83-84. In this section, anthropometric measurements included BMI and body weight at the same time. Since BMI was calculated from both weight (kg) and height (cm), I strongly advise you to retain BMI and remove the body weight in order to reduce redundancy. In addition, you can also choose to add the information of height (cm).
response 2: Body weight was removed to reduce redundancy.
Comment 3: Lines 126-127. Maybe this section should be revised: Statistically significance was set at two-sided P<0.05. In addition, I advise you to retain only two-sided test P values in Table 2.
Response 3: two-sided test p-values were retained in table 2. one-sided was eliminated.
Comment 4: One general comment on results: text often duplicates tables’ content, and the order of your results is a little confusing. I strongly advise you to simplify and adjust this section, as it complicates reading.
Response 4: Thank you for your feedback. I have revised the results section to address your concerns.
Comment 5: In Table 1. Hypertension was considered as a major determinant, SBP and DBP were also used to determine whether participants have hypertension. But there are other criteria for hypertension, such as take antihypertensive medications. So I suggest you to explain your diagnostic criteria for hypertension clearly in the Materials and Methods section.
Response 5: Thank you for your feedback. I have provided clarification on lines 93-94.
Comment 6: Table 2 was not standardized; it is recommended to remove the top and bottom border lines of each row. In addition, I advise you to delete the columns of One-sided P values in Table 2. You can refer to the format of Table 1.
Response 6: The top and bottom border lines of each row have been removed, and one sided p-values have been deleted from Table 2.
Comment 7: I agree with the content of this piece of discussion in general, especially with what is reported and particularly if it was in the introduction. However, there are some problems with it: Lines 172-175. Such a section is really too repeated with the Introduction, if I can advise, I would delete it. It is also possible to refine and summarize the content of your results. Furthermore, I suggest supplementing some content of possible mechanisms that could support your results in this study to the second paragraph of the Discussion section.
Response 7: I have revised the Discussion section to address your concerns. The content of the results has been refined and summarized for better clarity. Additionally, I have provided possible mechanisms that could support the study's results. Please review the updated section and let me know if any further adjustments are needed.
Comment 8: You did no add an appropriate strengths section, I suggest adding it in the Discussion section, which should include at least: a) study the information of male and female respondents separately; b) the participants were monitored comprehensively for 10 weeks, and so on.
Response 8: Thank you for your feedback. I have added a strengths section to the Discussion, addressing the points you mentioned on lines 234-264.
Reviewer 4 Report
Comments and Suggestions for Authors
The manuscript of King et al. explores the effects of exercises in T2DM patients on several parameters. The manuscript is well-presented and the authors described the limitation. Before, accepting I would like to suggest to the authors to include a table showing the pre and post diferenças regarding sex and the evaluated parameters.
Author Response
Dear Reviewer 4,
Comment 1: The manuscript of King et al. explores the effects of exercises in T2DM patients on several parameters. The manuscript is well-presented and the authors described the limitation. Before, accepting I would like to suggest to the authors to include a table showing the pre and post diferenças regarding sex and the evaluated parameters.
Response 1: Thank you for your positive feedback on our manuscript. We appreciate your suggestion to include a table showing the pre- and post-differences regarding sex and the evaluated parameters. The primary variable of HbA1c compared sex differences is already provided in Table 3 of the document. The other variables were considered for secondary analysis and were not included in the main tables to prevent confusion and maintain clarity. We believe this approach keeps the focus on the primary outcomes of the study. Thank you for your understanding.
Reviewer 5 Report
Comments and Suggestions for Authors
- The intervention was divided into a walking group and a non-walking group. Is it reasonable to further divide the groups by gender?
- In Table 1, the gender ratio within the walking and non-walking groups should be verified.
- For example, the male group (26 participants) includes both walking and non-walking participants. The same applies to the female group (41 participants).
- What analyses were performed to obtain the one-sided P and two-sided P values in Table 2?
- Please add the unit for functional reach in Table 2.
- Wouldn't it be more appropriate to analyze the male and female groups within the walking group separately? The same applies to the non-walking group.
- Since the objectives and analyses are not appropriate, please reanalyze and rewrite the discussion based on the new results.
Author Response
Dear Reviewer 5,
Comment 1: The intervention was divided into a walking group and a non-walking group. Is it reasonable to further divide the groups by gender?
Response 1: In Table 1, males and females are divided (with corresponding p-values), and walking and non-walking groups are divided (with corresponding p-values). We don’t feel it would be valuable to add four more columns to Table 1 (male walking, male non-walking, female walking, female non-walking), but we do address sex in section 3.1 and walking and non-walking in section 3.2.
Comment 2: In Table 1, the gender ratio within the walking and non-walking groups should be verified. For example, the male group (26 participants) includes both walking and non-walking participants. The same applies to the female group (41 participants).
Response 2: The non-walking group has 17 males (39.5%) and 26 females (60.5%). The walking group has 9 males (39.1%) and 14 females (60.9%). We have added this to the manuscript. Lines 143-145
Comment 3: What analyses were performed to obtain the one-sided P and two-sided P values in Table 2?
Response 3: Paired-samples t-test. This is stated in the name of the table (page 4, Line 146).
Comment 4: Please add the unit for functional reach in Table 2.
Response 4: unit for functional reach has been added in Table 2.
Comment 5: Wouldn't it be more appropriate to analyze the male and female groups within the walking group separately? The same applies to the non-walking group. Since the objectives and analyses are not appropriate, please reanalyze and rewrite the discussion based on the new results.
Response 5: We feel subdividing the sample into these four different subgroups would be inappropriate, as the subsamples would be too small for the analyses we are performing (e.g., there are only 9 males in the walking group, which would be inappropriate for regression), and too many statistical comparisons would be made; accordingly, our risk of both Type I and II errors would be substantial if we subdivided further.
Round 2
Reviewer 1 Report
Comments and Suggestions for Authors
Dear Authors,
The answers to some comments have not been seen in the revised version.
1. For the convenience of the readers, please draw a schematic figure of how to do the design and get the results.
2. Make a list of all existing abbreviations and put them at the beginning of the article.
Author Response
Comment 1: For the convenience of the readers, please draw a schematic figure of how to do the design and get the results.
Response 1: A figure has been added to the manuscript and can be found on lines 103-104.
Comment 2: Make a list of all existing abbreviations and put them at the beginning of the article.
Response 2: Abbreviation table has been added and can be found on lines 27-28
Reviewer 5 Report
Comments and Suggestions for Authors
Thank you for diligently addressing the comments I provided in the previous round.
I recommend that the revised manuscript be accepted for publication.
Author Response
Comment 1: Thank you for diligently addressing the comments I provided in the previous round.
I recommend that the revised manuscript be accepted for publication.
Response 1: Thank you for providing your recommendations for the publication.